# Preclinical Assessment of Immunogenicity and Protectivity of Novel ROR1 Fusion Proteins in a Mouse Tumor Model

**DOI:** 10.3390/cancers14235827

**Published:** 2022-11-26

**Authors:** Hadi Hassannia, Mohammad Mehdi Amiri, Mojgan Ghaedi, Ramezan-Ali Sharifian, Forough Golsaz-Shirazi, Mahmood Jeddi-Tehrani, Fazel Shokri

**Affiliations:** 1Immunogenetics Research Center, School of Medicine, Mazandaran University of Medical Sciences, Sari P.O. Box 48157-33971, Iran; 2Department of Immunology, School of Public Health, Tehran University of Medical Sciences, Tehran P.O. Box 14155-6559, Iran; 3Department of Hematology and Oncology, Imam Khomeini Hospital, Tehran University of Medical Sciences, Tehran P.O. Box 14197-33141, Iran; 4Monoclonal Antibody Research Center, Avicenna Research Institute, ACECR, Tehran P.O. Box 19839-69412, Iran

**Keywords:** ROR1, fusion proteins, vaccine, active immunotherapy

## Abstract

**Simple Summary:**

Receptor tyrosine kinase-like orphan receptor 1 (ROR1) is a tumor-associated antigen reported to be overexpressed in a variety of malignancies. The aim of this study is preclinical evaluation of various ROR1 fusion proteins as novel cancer vaccines in a fully syngeneic mouse tumor model. Our results showed that a fusion protein containing mouse ROR1, IgG Fc, and a universal tetanus toxin (TT) helper T-cell carrier could break tolerance against mouse ROR1 autoantigen and completely inhibit the growth of syngeneic ROR1+ tumor cells.

**Abstract:**

The receptor tyrosine kinase-like orphan receptor 1 (ROR1) is a new tumor associated antigen (TAA) which is overexpressed in several hematopoietic and solid malignancies. The present study aimed to produce and evaluate different fusion proteins of mouse ROR1 (mROR1) to enhance immunogenicity and protective efficacy of ROR1. Four ROR1 fusion proteins composed of extracellular region of mROR1, immunogenic fragments of TT as well as Fc region of mouse IgG2a were produced and employed to immunize Balb/C mice. Humoral and cellular immune responses and anti-tumor effects of these fusion proteins were evaluated using two different syngeneic murine ROR1+ tumor models. ROR1-specific antibodies were induced in all groups of mice. The levels of IFN-γ, IL-17 and IL-22 cytokines in culture supernatants of stimulated splenocytes were increased in all groups of immunized mice, particularly mice immunized with TT-mROR1-Fc fusion proteins. The frequency of ROR1-specific CTLs was higher in mice immunized with TT-mROR1-Fc fusion proteins. Finally, results of tumor challenge in immunized mice showed that immunization with TT-mROR1-Fc fusion proteins completely inhibited ROR1+ tumor cells growth in two different syngeneic tumor models until day 120 post tumor challenge. Our preclinical findings, for the first time, showed that our fusion proteins could be considered as a potential candidate vaccine for active immunotherapy of ROR1-expressing malignancies.

## 1. Introduction

Cancer is one of the leading causes of morbidity and mortality worldwide, with estimated 19.3 million new cases and almost 10 million cancer deaths in 2020 [1]. Within the last three years, the diagnosis and treatment of cancer was adversely affected by the coronavirus disease 2019 (COVID-19) pandemic [2]. Chemotherapy, radiotherapy, and surgical resection of tumors are currently the main therapeutic options for cancer therapy [3]. However, these approaches are not completely effective and disease relapse occurs in most cases [4]. In addition to conventional therapies, immunotherapy has been considered as a promising alternative approach to treat malignancies [5]. Recent progresses in cancer immunotherapy has considerably revolutionized the treatment of several cancers [6]. A large number of therapeutic monoclonal antibodies (MAbs) have so far been produced against a variety of TAAs and successfully employed in a number of malignancies [7]. By targeting TAAs and other relevant immunological targets, cancer immunotherapy strengthens the patients’ suppressed immune system to re-target and destroy cancer cells, which might have escaped surgery or have metastasized [8]. Despite the progress in treating cancer with therapeutic MAbs, a significant number of patients still die, indicating the need for more effective tools for cancer therapy. This approach has encountered several shortcomings, such as failure to induce immunological memory, short half-life of antibody, need for repeated injections to maintain protective immunity, resistance to treatment due to mutated tumor cells, not effective against intracellular targets and requirement for high expression level of target antigen on surface of tumor cells. Such drawbacks imply the need for alternative modalities such as active immunotherapy [9,10,11].

Active immunotherapy is another form of immunotherapy that attempts to stimulate the host immune system against tumor antigens to attack and eradicate the cancer cells [12]. The advantages of cancer vaccines over other therapies include durable protective immune response by induction of immunological memory, stimulation of both humoral and cellular immune responses and low toxicity [13,14]. However, immune tolerance to tumor self-antigens has been one of the major obstacles to the development of effective cancer active immunotherapy. A current challenge in active immunotherapy for cancers is breaking immune tolerance to tumor self-antigens [15]. Various strategies have been used to break self-tolerance, including modification of TAAs through conjugation to highly-immunogenic carrier proteins [16]. The carboxy-terminal fragment C (FrC) of tetanus toxin (TT), which is highly immunogenic, but non-toxic, is extensively used as a carrier in cancer vaccines [17]. In vaccine design, fusion of a target protein to the immunoglobin Fc domain is reported to increase the immunogenicity of vaccine antigens [18]. Tagging the proteins with immunoglobulin Fc domain extend the protein half-life by salvaging the protein of interest from endosomal degradation, improve stability and solubility, and also increase the expression and secretion of the protein as well as simplifies the purification of the protein [19]. The Fc-fusion proteins can strongly trigger humoral immune responses by improving the uptake of antigens by antigen presenting cells through Fc-FcγR interactions as well as, induction of cross-presentation via the MHC class I and II pathways [19].

Careful selection of TAA is critical for specificity and effectiveness of cancer immunotherapy to induce tumor-specific killing with minimal damage to normal cells.

The oncofetal protein, receptor tyrosine kinase-like orphan receptor 1 (ROR1), is considered as a promising target for cancer therapy due to its overexpression on many types of tumor cells with low expression in normal adult tissues [20]. ROR1-expressing hematologic and solid tumors have a high potential for self-renewal, exhibit increased survival and migration, and are associated with poor outcomes [21]. So far, several therapeutic strategies including mAbs, bispecific mAbs, CAR-T cells and small molecule inhibitors against ROR1 have been evaluated and showed promising results in preclinical and clinical trials, but none of them have been approved for clinical use [20].

It was suggested that ROR1 is a common autoantigen in cancer patients [22]. However, ROR-1 as a self TAA displays poor immunogenicity and cannot elicit a robust immune response [23,24]. Here, by developing a syngeneic mouse ROR1 (mROR1) tumor model [25], we investigated the immunogenicity of mROR1, as a murine TAA, and four different ROR1 fusion proteins containing different carriers and tags and also assessed the protective efficacy of the fusion proteins as anti-cancer candidate vaccine.

## 2. Materials and Methods

### 2.1. Cell Lines and Mice

Highly tumorigenic mouse cell lines stably expressing full-length mROR1 protein were generated from CT26 (colon carcinoma) and 4T1 (breast carcinoma) cell lines (National Cell Bank of Iran, Pasteur Institute, Tehran, Iran) as described previously [25]. Tumor cells were cultured in complete medium including RPMI1640 medium supplemented with 10% heat-inactivated fetal bovine serum and 100 U/mL penicillin/streptomycin (Gibco Life Technologies, Paisley, UK), and incubated at 37 °C in a humidified atmosphere of 5% CO2. Six- to eight-week-old female BALB/c mice were purchased from Pasteur Institute of Iran and housed in animal facility. Animal use protocol were in accordance with guidelines approved by Tehran University of Medical Sciences.

### 2.2. Construction, Expression, and Purification of ROR1 Fusion Proteins

In this study, to break immunologic tolerance to a self TAA (mROR1 protein), Pan Th epitopes from tetanus toxin, which binds to the majority of MHCII alleles in mouse and human; primary domain of fragment C (p.Dom) and P2-P30 fused to extracellular region of mROR1 TAA with a flexible linker accompanied Fc fragment of mouse IgG2a. The order of each segment and abbreviation names of expression constructs is shown in Figure 1A. The gene coding of each segment is as follow: C-terminus of tetanus toxin (p.Dom, UniProt accession no. P04958), p2-p30 (QYIKANSKFIGITEL-FNFSFWLRVPKVSASHLE), flexible linker consisting of four glycines and a serine, extracellular domain of mouse ROR1 (mROR1-ECD; UniProt Q9Z139), and Fc fragment of mouse IgG2a (hinge region and CH2-CH3 domains; UniProt P01863). Each construct was commercially synthesized (Biomatik, Cambridge, ON, Canada) and inserted into a homemade plasmid that contains glutamine synthetase (GS) gene as a selection marker. The final constructs were then transfected to CHO-K1 cells by Lipofectamin 3000 reagent (Lifetechnology, Waltham, MA, USA) and stable clones were selected in the presence of methionine sulfoximine (MSX, Sigma-Aldrich, Steinheim, Germany). The final selected clones with the highest expression level of recombinant mROR1 fusion proteins were selected and adapted to growth in serum-free medium, (SFM), First CHOice^®^ Medium (UGA Biopharma, Hennigsdorf, Germany). The supernatants were collected and stored at −20 °C until purification. The fusion proteins were purified by protein G (M17, H17, B11) and Ni-NTA (X12) columns [26].

As described previously [25], the reactivity of mROR1 fusion proteins examined by rabbit anti-mROR1 pAb using ELISA method. The purity and integrity of ROR1 fusion proteins were determined by SDS-PAGE.

### 2.3. Mouse Vaccination and Sample Collection

For prophylaxis studies, one hundred and forty-four Balb/C mice were randomly divided into eight groups (18 mice per group). Four groups were immunized with 25 μg per mouse of fusion proteins M17, H17, B11 and X12 emulsified in complete Freund’s adjuvant (Sigma, St. Louis, MO, USA) for the first injection and then boosted 4 times biweekly with 12.5 μg of the same fusion proteins emulsified in incomplete Freund’s adjuvant (Sigma). One group was immunized with 12.5 μg of H17 plus 10 μg of CpG (TCGTCGTTTTGTCGTTTGTCGTT) (Bioneer, Daejeon, Republic of Korea) for five times. Three control groups were injected with Freund’s adjuvant, CpG or PBS. Blood samples were collected before each injection and two weeks after the last immunization and sera were separated and frozen for later use. Seven mice of each group were sacrificed to investigate humoral and cellular immune responses and the tumor analysis was performed on the remaining mice. A schematic of the vaccination and analysis timeline is presented in Figure 2A.

#### 2.3.1. Humoral Responses to Fusion Proteins

The antibody response against different fusion proteins and ROR1-specific antibody in the sera of immunized mice were measured by indirect ELISA. Briefly, 5 µg/mL mROR-Fc protein or different fusion proteins were coated on 96-well ELISA plates (Nunc Maxisorp, Roskilde, Denmark) and incubated at 4 °C overnight. After blocking with skim milk 5% (in PBS with 0.05% Tween20 (Sigma-Aldrich, Steinheim, Germany)), mouse sera were titrated. Next, HRP-conjugated goat anti-mouse F(ab’)2 (Abcam, Cambridge, UK) was diluted in blocking buffer, added to the wells and incubated at 37 °C for an hour. Plates were then washed and tetramethylbenzidine (TMB) (Pishtazteb, Tehran, Iran) substrate was added. The reaction was stopped by 0.5M HCL and the optical density (OD) was measured at 450/600 nm by ELISA reader (BioTek, Winooski, VT, USA).

#### 2.3.2. Cytokine Responses to Fusion Proteins

To investigate the dominant type of T helper immune response to different fusion proteins, the level of interferon-γ (IFN-γ), IL-4, IL-17, and IL-22 was measured in the culture media of the splenocytes. Two weeks after the last immunization, splenocytes of the immunized animals were cultured at 2 × 10^6^ cells/well in 24-well cell culture plates (JET BIOFIL, Guangdong, China). The cells were then stimulated with 10 μg/mL of the immunizing fusion proteins, and mROR1-Fc for 72 h. Untreated cells served as negative controls. Supernatants of stimulated and unstimulated cells were collected and stored at −80 °C until cytokine measurement. Cytokines were measured by commercial sandwich ELISA kits according to the manufacturer’s instructions (eBioscience, Affymetrix, CA, USA). The fold change of cytokines was reported relative to the control value for unstimulated controls.

#### 2.3.3. Frequency of Cytotoxic T Lymphocyte Responding to ROR1+ Tumor Cells

We further studied whether the fusion protein vaccines evoke a cytotoxic CTL response. For enumeration of CD8^+^ CD107^+^ cells, 10^6^ splenic cells obtained from immunized and adjuvant control mice were co-cultured with 2 × 10^5^ ROR1-positive syngeneic tumor cells in the presence of allophycocyanin (APC) conjugated CD107-antibody and 10 μmol/L monensin (Biolegend, San Diego, CA, USA) at 37 °C for 6 h [27]. Cells were then washed and stained with FITC-conjugated anti-CD8 antibody and scanned by a flow cytometer (Partec, Nuremberg, Germany), and data were analyzed using the FlowJo v10 software (Tree Star, Ashland, OR, USA).

### 2.4. Assessment of the Protective Efficacy of the Fusion Proteins in Mouse Tumor Models

For in vivo tumor protection experiments, two weeks after the last immunization, the rest of vaccinated Balb/C mice were challenged with fully syngeneic CT26-ROR1+ and 4T1-ROR1+ tumor cells (The experimental design for tumor challenge is shown in Figure 3A). To establish the 4T1- and CT26-ROR1+ tumor model, 1 × 10^6^ CT26-ROR1+ or 5 × 10^5^ 4T1-ROR1+ tumor cells were injected subcutaneously into the flanks and mammary fat pads of the immunized Balb/C mice, respectively [25]. In 24 days after tumor inoculation, 5 mice in each group from CT26-ROR1+ tumor challenged mice were sacrificed and their splenocyte were assessed by flow cytometry for measuring the immunosuppressive cell population, including Gr1^+^ CD11b^+^ MDSC and CD4^+^ Foxp3^+^ Treg. Fluorescent labeled antibodies directed against the following markers and also their isotype-matched control were obtained from Biolegend (San Diego, CA, USA): Gr1( APC conjugated), CD11b (PE conjugated), CD4 (APC conjugated), and FoxP3 (Alexa Fluor 488 conjuaged). Each antibody was titrated using serial dilutions and the optimal concentration was determined. 10^6^ splenocyte cells were incubated with an optimized concentration of fluorochrome-conjugated mAbs at 4 °C for 30 min. FoxP3 intracellular staining was performed by Cytofix/Cytoperm Fixation/Permeabilization kit (BD Biosciences, San Diego, CA). After washing twice, stained cells were evaluated using a flow cytometer (Partec, Nuremberg, Germany), and data were analyzed using the FlowJo v10 software. The tumor size of each animal was measured every 2 days using a digital caliper and tumor volumes were followed for up to 60 days. Survival end point was set when tumor volume reached about 3500 mm^3^ or the time to death.

### 2.5. Statistical Analysis

Statistical analyses were performed using SPSS statistical package (SPSS, Chicago, IL, USA) and GraphPad Prism (San Diego, CA, USA). Two group comparisons were assessed with a Mann-Whitney U test. Comparisons between groups were investigated by one-way ANOVA analysis. Kaplan-Meier survival curves were created and the differences between the curves were analyzed by log-rank test. Spearman’s non-parametric correlations were used to determine relationships between variables. Data are presented as a mean ± standard deviation. * *p* < 0.05, ** *p* < 0.01, *** *p* < 0.001, and **** *p* < 0.0001 were considered statistically significant difference.

## 3. Results

### 3.1. Production and Characterization of Fusion Proteins

We successfully produced mROR1 fusion proteins in eukaryotic host cells and purified them. The schematic diagram representing the structure of different constructs and fusion proteins is shown in Figure 1A,B. All purified fusion proteins were characterized by ELISA and SDS-PAGE. As shown in Figure 1C, the reactivity of Fc fusion proteins (M17, H17, B11) with anti-mROR1 is higher than that of non-Fc fusion protein (X12), which might be due to the effect of Fc domain on mediating the correct folding or dimerization of the fusion proteins. Moreover, the purity and structure of fusion proteins were analyzed by SDS-PAGE under reducing and non-reducing conditions and the corresponding bands of M17 at ~220 kDa, H17 at ~190 kDa, B11 at ~170 kDa and the monomer band of X12 at ~63 kDa under non-reduced condition were detected (Figure 1D).

### 3.2. ROR1 Fusion Proteins Induced Potent mROR-1specific Immune Responses in Immunized Mice

Five doses of the purified fusion proteins were administered to Balb/C mice. Two weeks after the last dose immunization, seven mice of each group were sacrificed to investigate humoral and cellular immune responses (Figure 2A). For evaluation of the antibody response to fusion proteins and mROR1, serum samples were collected and measured by ELISA. Our data showed that ROR1 fusion proteins were able to induce potent mROR1 specific antibody response in immunized mice (Figure 2B,C) compared to adjuvant control groups. In addition, fusion proteins containing both TT pan-Th epitopes as well as Fc fragment (M17 and H17) induced higher levels of ROR1-specific antibody compared to fusion proteins containing either TT domain (X12) or Fc fragment (B11) alone. Comparison of the ROR1-specific antibody levels between groups indicates superiority of the H17 fusion protein in combination with different adjuvants. It is noteworthy to mention that immunization with a large protein carrier (p.Dom in M17 fusion protein) induced carrier-specific antibody. As expected, antibody response against Fc protein was negligible in all groups compared to the adjuvant control group.

### 3.3. TT Carrier Peptides Mediate a Strong Cytokine Response against Fusion Proteins

To further investigate the pattern of Th response to different fusion proteins, spelenocytes of the immunized mice were isolated and after restimulation with their corresponding immunogens and B11 (mROR1-Fc), the levels of IFN-γ, IL-4, IL-17 and IL-22 were measured in their supernatants. As shown in Figure 2D, all immunizing proteins were able to induce a significant IFN-γ response against the corresponding immunogens and mROR1-Fc in comparison to the control groups. However, immunization of the mice with the fusion proteins containing TT carrier (M17, H17 and X12) induced a higher IFN-γ response compared with the fusion protein lacking the TT carrier peptide (TAA + Fc; B12). In addition, a significant increase of IL-4 was induced only in mice immunized with M17 fusion protein, suggesting that p.Dom is responsible for induction of this cytokine (Figure 2E). As shown in Figure 2F the IL-17 cytokine response was significantly induced by groups of mice immunized with fusion proteins containing TT carrier (M17, H17 and X12). Similar to IFN-γ, all formulated immunizing proteins were able to induce an IL-22 response against the corresponding fusion proteins as well as mROR1-Fc in comparison to the control groups. Generally, no cytokine response was detected against the Fc fragment in all groups of immunized mice. The pattern of cytokine response was similar in mice immunized with H17 fusion protein in combination with either IFA or CpG adjuvants. We also analyzed the statistical correlation between the antibody titers and cytokine responses. As shown in Figure 2H, no significant correlation was observed between mROR1-specific antibody titers and mROR1-induced cytokine levels in all groups of immunized mice. However, there was a significant positive correlation between levels of mROR1-specific antibody and carrier induced IFN-γ (r = 0.750, *p* < 0.001), IL-17 (r = 0.552, *p* < 0.001) and IL-22 (r = 0.743, *p* < 0.001) cytokine responses (Figure 2I) which indicates cross helping of ROR1-specific-B cells by carrier-specific T cells.

### 3.4. TT Peptides Formulated Fusion Proteins Induce Potent TCD8+ Response against ROR1+ Tumor Cells

CD8+ T cells are essential for the anti-tumor effect. Hence, we studied whether the fusion proteins are able to evoke a cytotoxic CTL response. Initially, the tumor cells were co-cultured with splenic cells obtained from immunized mice (Figure 2J, upper panel). Then, flow cytometric analysis was performed to determine the percentage of CD8+ T cells which express surface CD107 as a marker for activated CTL cells (Figure 2J, lower panel). Our results showed that immunization of the mice with fusion proteins containing both the TT peptides and Fc fragment induced a potent anti-tumor CTL response against ROR1+ tumor cells (Figure 2K). We also analyzed correlation between the TAA specific CTL response and TAA or carrier induced cytokine responses. As shown in Figure 2L, no significant correlation was observed between these two parameters. However, there was a significant positive correlation between levels of mROR1-specific CTL frequency and carrier induced IFN-γ (r = 0.707, *p* < 0.001), IL-4 (r = 0.402, *p* < 0.05), IL-17 (r = 0.501, *p* < 0.002) and IL-22 (r = 0.674, *p* < 0.001) cytokine responses (Figure 2M).

### 3.5. Immunization with Fusion Proteins Inhibits Tumor Growth in Mice Implanted with Syngeneic ROR1+ Tumor Cells

To evaluate the inhibitory effect of mROR1 fusion proteins on tumor growth in vivo, we inoculated the vaccinated mice with fully syngeneic CT26-ROR1+ (8 mice in each group) and 4T1-ROR1+ tumor cells (3 mice in each group). The experimental design for tumor challenge is shown in Figure 3A. The results of CT26-ROR1+ tumor challenge indicate that the mean tumor size in mice immunized with the TT peptide and Fc fragment containing fusion proteins (M17, H17/IFA and H17/CpG) was significantly less than that of the control group (*p* < 0.0001) (Figure 3B). Both IFA and CpG adjuvants induced a similar response in the H17 group. In addition, a significant tumor protection was observed in mice immunized with TT + TAA (X12) fusion protein (*p* < 0.05). However, immunization of mice with TAA + Fc (B11) delayed tumor growth, without significant tumor growth inhibition compared to the control group. Follow up on all groups of mice up to 120 days post tumor inoculation showed prolonged survival in mice immunized with the TT peptide and Fc fragment containing fusion proteins (Figure 3C). Similar data was obtained in mice implanted with 4T1-ROR1+ tumor cells (Figure 3D,E).

### 3.6. Immunization with ROR1+ Fusion Proteins Decreased the Frequencies of Treg and MDSC Populations

Also, we determined the frequency of Treg and MDSC cells in spleen cells of immunized mice 24 days post CT26-ROR1+ tumor induction by flow cytometry. Representative dot plots indicating the analysis method used for enumeration of the cells are shown in Figure 3F. Immunization with all fusion proteins resulted in lower frequencies of both Treg and MDSC populations, (Figure 3G,H), though this decrease was more significant in mice immunized with the TT peptide and Fc fragment containing fusion proteins. The frequencies of both cell populations displayed a strong positive correlation with tumor size (*p* < 0.0001) (Figure 3I).

## 4. Discussion

Vaccination is one of the most successful approaches for preventing infectious diseases [28]. The role of vaccination in the treatment of non-infectious diseases such as cancer and allergic diseases has gained increasing attentions in recent years [29]. However, due to the selfness origin of the tumor antigens, breaking tolerance to these antigens is a major challenge to induce an efficient anti-tumor immune response [30]. Hence, it is highly necessary to develop an efficient strategy to overcome the self-protein tolerance and promote potent immunity against tumor vaccines. In this study, to break immunologic tolerance to a self TAA (mROR1 protein), we designed and produced five different recombinant fusion proteins of mROR1 (Figure 1) and investigated their immunogenicity and protective efficacy as an anti-cancer candidate vaccine in fully syngeneic ROR1+ mouse tumor models. Despite many interests in use of ROR1 molecule as a target for cancer active therapy, so far it has not been introduced in syngeneic ROR1+ animal model and several studies have commonly used human tumor xenografts model in immunosuppressive microenvironments [23,31]. These models are not only costly and laborious, but also the results of these models may not actually simulate human tumor models [32]. Hence, another approach is to develop syngeneic mouse ROR1+ tumor models and assess the anti-tumor effects of active immunization in mice.

The whole extracellular domain of ROR1 was used for construction of fusion proteins as they contain a broadly diverse repertoire of epitopes which could be presented by a variety of both MHC I and II molecules. In addition, due to the presence of known immunogenic MHC class I binding epitopes in the second domain of FrC, which could compete with weaker tumor-derived epitopes by the phenomenon of immunodominance, we selected p.Dom and p2-p30 sequences as a carrier, but not full FrC. Our results of humoral immune response showed that the level of antibody in groups of mice immunized with fusion proteins containing the TT carrier, especially P2-P30, along with Fc (H17) was significantly higher than that in groups immunized with fusion proteins lacking the TT carrier or the Fc fragment (Figure 1C). Considering the fact that an ideal carrier protein does not elicit an immune response to itself, we found that immunization with a large protein carrier (p.Dom in M17 fusion protein) induced high levels of carrier-specific antibody which was accompanied with induction of significant levels of IL-4 (Figure 2E). Therefore, further reduction of the size of carrier domain to short helper epitopes is expected to induce a narrower and more specific antibody response against the target TAA [33].

The cytokine assay results showed that the Th response was largely induced against the TT carrier epitopes, rather than ROR1, and splenocytes from mice vaccinated with TAA + carrier + Fc fusion proteins actively secreted IFN-γ, IL-17 and IL-22. In accordance with our study, results of other studies using PADRE or P2 and/or P30 epitopes of TT as carriers in DNA-based β-Amyloid (Aβ) Alzheimer’s disease vaccines [34,35,36] showed that, the vaccine induced Aβ specific humoral immune responses in mice and also strong anti-P2 and anti-P30 Th-cell responses [36]. We also observed a significant positive correlation between the TAA antibody levels and carrier induced IFN-γ, IL-17 and IL-22 cytokine responses (Figure 2I). This indicates that mROR1-specific B cells bind to the antigen through the mROR1 determinants, facilitating the endocytose of the TAA-carrier fusion proteins, and present peptides derived from the carrier protein to carrier-specific helper T lymphocytes. Therefore, the two cooperating lymphocytes recognize two different epitopes of the same carrier-antigen complex (Figure 4). Similarly, the Fc protein seems also to contribute to the efficient reactivity of the mROR1-specific B cells, which explains why induction of cytokines to the carrier peptides was more significant in fusion proteins containing both the TT and Fc proteins than that lacking the Fc fragment (X12) in immunized mice (Figure 2D,F,G). Similarly, in different studies have been shown that the targeting immunogens to Fcγ receptors through Fc region could selectively facilitate uptake of antigens and elevate humoral and cellular immune response [37,38].

Regarding the CTL responses, our results showed that mROR1-specific CTL cells, were more expanded in mice immunized with carrier-TT along with Fc fragment fusion proteins (M17 and H17). Previous studies have indicated that the help delivered through B cells as well as DCs, which are able to engage the antigen with FcR and are dominant cross-presenting cells in many systems, are critical components to activate CTL responses [39]. Moreover, our results showed that reactivity of mROR1-Fc fusion proteins (M17, H17, B11) with anti-mROR1 polyclonal antibody were higher than tag free mROR1fusion protein, i.e., X12 (Figure 2C). This could be due to the Fc mediated dimerization and/or correct folding of ROR1, because no reactivity to the mouse Fc was detected with this antibody. In accordance with our findings, several studies have indicated that Fc fragment may enhance immunogenicity through extending half-life of the antigen, enhancing its correct folding, increasing the size of antigen by dimerization, Fc receptor-mediated uptake by antigen-presenting cells, induction of cross-presentation and efficiently cross-priming CD8 T cells via CD4 T cells help, addressing FDC to trap Ag-IgG Fc proteins in a complement-independent manner and increasing expression of the antigen [19].

Results of the tumor challenge experiments showed that M17 and H17 fusion proteins (carrier + TAA + Fc), regardless of the adjuvant type, inhibited growth of the ROR1+ tumor cells more efficiently than the other fusion proteins (Figure 3B,D). In addition, M17 and H17 fusion proteins potently increased survival time of tumor bearing mice compared to the control mice (Figure 3C,E). The tumor mass in mice vaccinated with M17 and H17 proteins, was completely regressed up to 120 days post tumor implantation and no recurrence of tumor was observed in these mice. Comparison of the present results of active immunotherapy with our previous results using passive immunotherapy [25], implies the superiority of active immunotherapy with carrier + TAA + Fc fusion proteins. In addition, inhibition of the tumor cells growth was accompanied by significant reduction of MDSC and Treg cells frequencies in tumor-bearing mice and the lowest frequency of the immunosuppressive cells was observed after vaccination with carrier + TAA + Fc proteins (Figure 3G,H). Here, we also observed a significant positive correlation between the tumor volume and frequency of immunosuppressive cells (Figure 3I). Moreover, we compared ROR1-specific immune responses and tumor growth inhibition induced by H17 fusion protein in combination with two different adjuvants, including Freund’s and CpG. The results revealed that both adjuvants have similar effect in stimulating ROR1-specific humoral and cellular mediated immunity (Figure 2 and Figure 3). This finding does not support our recent data, showing superiority of CpG over other adjuvants such as Freund’s, Montanide and Alum in a HER2 positive mouse tumor model [40]. This discrepancy might be due to the presence of Fc fragment in our H17 fusion protein which may strength the anti-tumor response to a level that superiority of the CpG adjuvant is no longer observed [18,19,41]. The type of the tumor may also contribute to this discrepancy because in the present study we employed syngeneic ROR1-expressing mouse colon and breast cancer cell lines, whereas in our previous study HER2-expressing rat breast tumor cell line was used [40].

Our results imply that the mechanisms of the anti-tumor effects of our fusion proteins are mediated through effector cells, including antigen specific B cells, Th cells, and CTLs (Figure 4).

## 5. Conclusions

Our preclinical data indicated that immunization with fusion protein containing universal p.DOM or P2-30 TT carrier sequences accompanied with the IgG Fc fragment significantly enhances the immunogenicity of a self TAA leading to inhibition of tumor cells growth and survival of the tumor bearing mice. These fusion proteins represent a promising and novel anticancer active immunotherapy strategy for targeting ROR1 expressing human cancers.

## Figures and Tables

**Figure 1 cancers-14-05827-f001:**
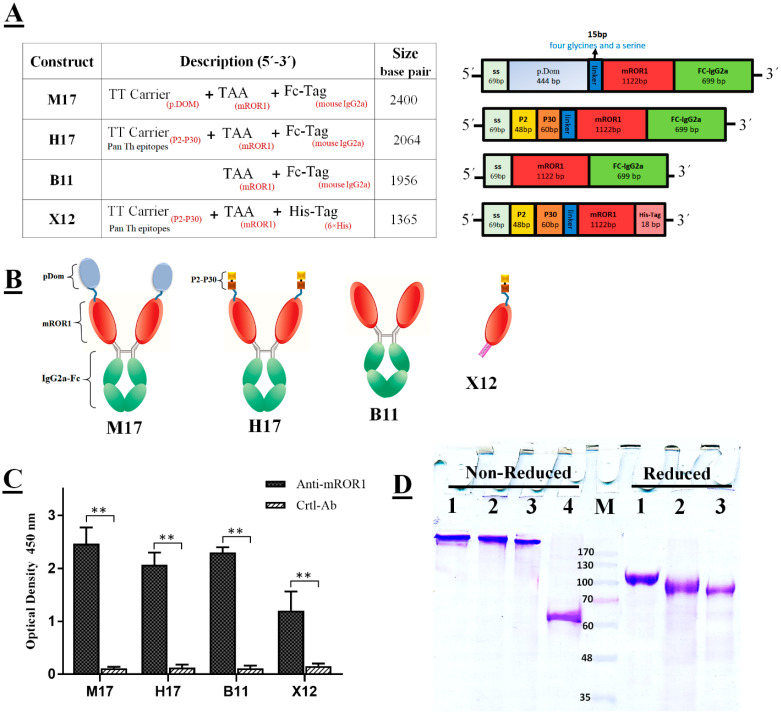
Design, production and characterization of ROR1 fusion proteins. (**A**) Detailed structure of expression constructs: Pan Th epitopes from tetanus toxin; p.Dom, P2, and P30 fused to mROR1 TAA with a flexible linker accompanied Fc fragment of mouse IgG2a. (**B**) Schematic illustration of the fusion proteins. (**C**) Assessment of reactivity of rabbit anti-mROR1 with each fusion protein. Normal (non-immune) rabbit serum was employed as a control. (**D**) SDS–PAGE analysis of each fusion protein under non-reducing (left panel) and reducing (right panel) conditions; M17 fusion protein (lane 1), H17 fusion protein (lane 2), B11 fusion protein (lane 3) and X12 fusion protein (lane 4). M represents molecular mass markers, with values in kilodaltons (KDa). (* *p* < 0.05 and ** *p* < 0.01).

**Figure 2 cancers-14-05827-f002:**
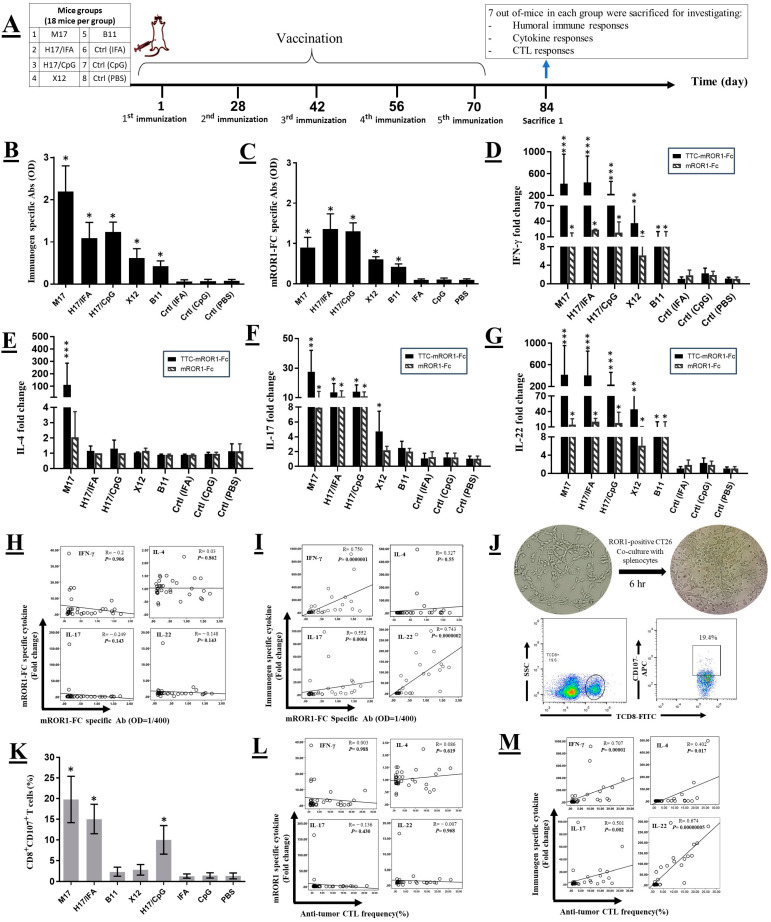
Assessment of immunogenicity of ROR1 fusion proteins. (**A**) Scheme of mice groups, immunization and assays timeline. Fusion proteins were administered subcutaneously at 5-time intervals. Two weeks after the fifth dose administration, seven mice of each group were sacrificed to investigate humoral and cellular immune responses. (**B**) Antibody against the immunizing protein and (**C**) mROR1 was assessed in serum 2 weeks after last immunization in all groups of mice. (**D**) Measurement of IFN-γ, (**E**) IL-4, (**F**) IL-17 and (**G**) IL-22 in culture supernatant of splenocytes from different groups of mice. (**H**) Correlation between mROR1 specific antibody levels in immunized mice and mROR1 induced cytokine responses. (**I**) Correlation between mROR1 specific antibody levels in immunized mice and carrier induced cytokine responses. (**J**) CTL lysosomal granule exocytosis was determined by CD8^+^ CD107^+^ splenocytes after co-culture with ROR1-positive syngeneic tumor cells. Representative dot plots demonstrating the analysis method for enumeration of CD107 + CD8+ cells. (**K**) Frequency of CD8 + CD107+ CTL cells responding to ROR1+ tumor cells. (**L**) Correlation between CTL frequency and mROR1 induced cytokine responses. (**M**) Correlation between CTL frequency and carrier induced cytokine responses. The results represent mean ± SD. The *p* values were calculated by Mann-Whitney U test in comparison to the control adjuvant groups (* *p* < 0.05, ** *p* < 0.01, and *** *p* < 0.001).

**Figure 3 cancers-14-05827-f003:**
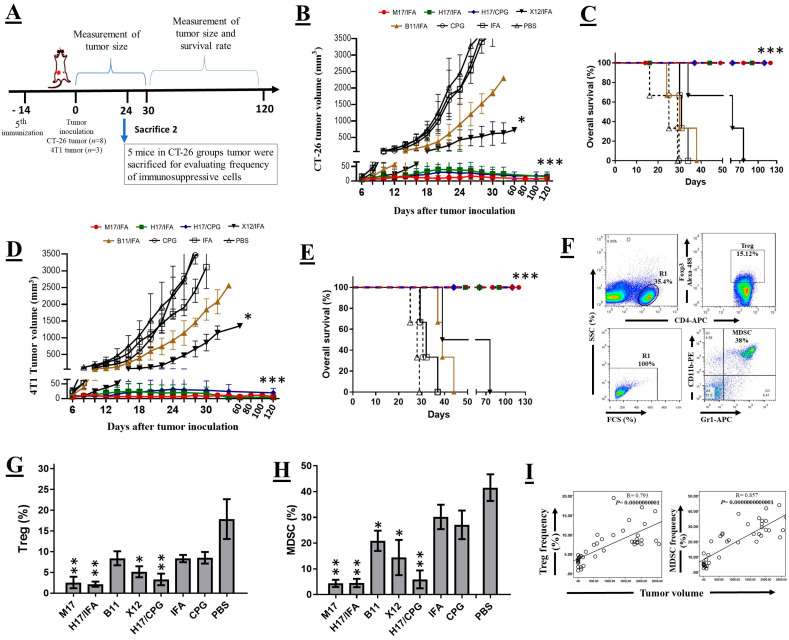
Protective efficacy of fusion proteins. (**A**) The experimental design for tumor challenge. (**B**) The growth curve of the CT26-ROR1 tumor after tumor challenge. (**C**) Kaplan–Meier survival curves of mice in each group after CT26-ROR1 tumor challenge. (**D**) The growth curve of the 4T1-ROR1 tumor. (**E**) Kaplan–Meier survival curves of mice in each group after 4T1-ROR1 tumor challenge. (**F**) Representative dot plots demonstrating the analyzing method used for enumeration of MDSC (below panel) and Treg (top panel). (**G**) Frequencies of MDSC and (**H**) Treg cells in the spleen of each group. (**I**) Correlation of Treg (left) and MDSC (right) frequencies with tumor volume. The results represent mean ± SD. The *p* values were calculated in comparison to the control adjuvant group (* *p* < 0.05, ** *p* < 0.01, and *** *p* < 0.001).

**Figure 4 cancers-14-05827-f004:**
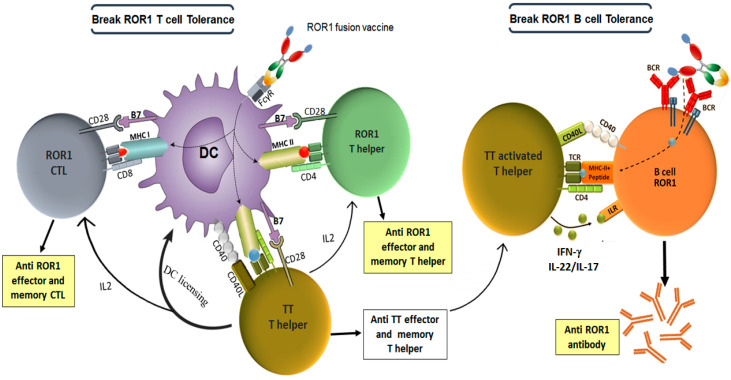
Proposed mechanism of action of TT-ROR1-Fc fusion proteins for activation of the humoral and cellular immune responses. (Left panel) pathways of provision of TT-specific CD4 T cell for DC maturation and breaking ROR1 T cell tolerance: main property of the ROR1 Fc-fusion protein is its ability to bind Fcγ receptors (FcγR), subsequently proteins are internalized by DC. This type of endocytosis promotes cross-presentation via MHC class I and II molecules, thereby priming both CD4 and CD8 T-cell responses. At first TT-specific CD4 T cells activated by the fusion protein, secrete IL-2, pro-inflammatory cytokines, and further activate the DC licensing, promoting the development of ROR1 specific T CD4/T CD8 via increasing costimulatory molecules on the DC (such as B7) and secreting cytokines. (Right panel) pathways of provision of TT-specific CD4 T cell help for ROR1-B cells: B lymphocytes recognize the ROR1 TAA through their BCR. BCR signaling and cytokines secreted by activated TT-specific CD4 T cells results in B cell activation and differentiation into ROR1 antibody producing plasma cells.

## Data Availability

The data that support the findings of this study are available from the corresponding author upon reasonable request.

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
