# Peer review of "Preclinical Assessment of Immunogenicity and Protectivity of Novel ROR1 Fusion Proteins in a Mouse Tumor Model"

_cancers, 2022, doi:10.3390/cancers14235827_

Round 1
Reviewer 1 Report
The manuscript describes the pre-clinical evaluation of four ROR-1 fusion proteins with immunogenic epitopes of TT and IgG2a Fc.
The study is well designed and executed with detailed description of methods. The discussion is well written with all the aspects of the study taken into consideration.
The manuscript can be accepted for the publication with some minor suggestions.
More Discussion on folding of these fusion proteins and impact of that on immunogenicity would add more significance to the paper.
References are missing at few places for e.g Page 2, Line 80.
Overall, the manuscript is well written and the work is very impressive.
Author Response
Comments and Suggestions for Authors
The manuscript describes the pre-clinical evaluation of four ROR-1 fusion proteins with immunogenic epitopes of TT and IgG2a Fc.
The study is well designed and executed with detailed description of methods. The discussion is well written with all the aspects of the study taken into consideration.
The manuscript can be accepted for the publication with some minor suggestions.
More Discussion on folding of these fusion proteins and impact of that on immunogenicity would add more significance to the paper.
The following statement was added “Similarly, in different studies have been shown that the targeting immunogens to Fcγ receptors through Fc region could selectively increase uptake of antigens and elevate humoral and cellular immune response” to the MS at page 11, lines 385-387. Moreover, the relation between proper folding and immunogenicity has already been discussed in page 11, lines 395-403 and 390-392.
References are missing at few places for e.g Page 2, Line 80.
The references were added at page 2, lines 50, 53, 56, 63 and 93 and page 12, line 425.
Overall, the manuscript is well written and the work is very impressive.
Reviewer 2 Report
The present study aims to design and produce five recombinant fusion proteins of mouse receptor tyrosine kinase-like orphan receptor 1 (ROR1) to break immunologic tolerance to a self TAA (mROR1 protein). The authors investigate the protective efficacy as an anti-cancer candidate vaccine. Tumor challenge in immunized mice shows that immunization with 34 TT (tetanus toxin)-mROR1-Fc fusion proteins inhibits tumor cells growth in tumor models until day 120 post tumor challenge. Altogether the data suggest that these fusion proteins are potential candidate vaccine for active immunotherapy of ROR1-expressing malignancies.
Specific comments.
- Comparison of the active immunotherapy data with previous results using passive immunotherapy is a key aspect of the study. A brief summary of the passive immunotherapy data would facilitate the reading and contribute to the impact of the study.
- Both Freund's and CpG adjuvants have similar effect in stimulating ROR1-specific humoral and cellular mediated immunity. This finding contradicts recent data showing superiority of CpG over other adjuvants. This discrepancy needs to be clarified
- Fc fragment enhances immunogenicity but the reasons explaining such an effect are unclear. Several possibilities are proposed like extending half-life of the antigen, enhancing its correct folding, increasing the size of antigen by dimerization. Which one should be priviledged.
- It has been suggested that treatment with ROR1 small interfering RNA (siRNA) can potentially mitigate invasive properties of cancerous cells and reverse the progression of cancer. What are the advantages of using fusion proteins.
Author Response
Comments and Suggestions for Authors
The present study aims to design and produce five recombinant fusion proteins of mouse receptor tyrosine kinase-like orphan receptor 1 (ROR1) to break immunologic tolerance to a self TAA (mROR1 protein). The authors investigate the protective efficacy as an anti-cancer candidate vaccine. Tumor challenge in immunized mice shows that immunization with 34 TT (tetanus toxin)-mROR1-Fc fusion proteins inhibits tumor cells growth in tumor models until day 120 post tumor challenge. Altogether the data suggest that these fusion proteins are potential candidate vaccine for active immunotherapy of ROR1-expressing malignancies.
Specific comments.
- Comparison of the active immunotherapy data with previous results using passive immunotherapy is a key aspect of the study. A brief summary of the passive immunotherapy data would facilitate the reading and contribute to the impact of the study.
The following statement was added “So far, several therapeutic strategies including mAbs, bispecific mAbs, CAR-T cells and small molecule inhibitors against ROR1 have been evaluated and showed promising results in preclinical and clinical trials, but none of them have been approved for clinical use” was added to the MS at page 2, lines 90-93.
- Both Freund's and CpG adjuvants have similar effect in stimulating ROR1-specific humoral and cellular mediated immunity. This finding contradicts recent data showing superiority of CpG over other adjuvants. This discrepancy needs to be clarified.
We have already mentioned in the discussion at page 12, lines 423-425 that this discrepancy might be due to the presence of Fc fragment in our H17 fusion protein which may strengthen the anti-tumor response to a level that superiority of the CpG adjuvant is no longer observed. The type of the tumor may also contribute to this discrepancy because in the present study we employed syngeneic ROR1-expressing mouse colon and breast cancer cell lines, whereas in our previous study HER2-expressing rat breast tumor cell line was used.
These descriptions have now been amended to the discussion at page 12 lines 424-430.
- Fc fragment enhances immunogenicity but the reasons explaining such an effect are unclear. Several possibilities are proposed like extending half-life of the antigen, enhancing its correct folding, increasing the size of antigen by dimerization. Which one should be priviledged.
Evidence in support of all these mechanism is documented, which has been explored in the review paper (ref 19). It is hard to choose one of them as the most important one.
- It has been suggested that treatment with ROR1 small interfering RNA (siRNA) can potentially mitigate invasive properties of cancerous cells and reverse the progression of cancer. What are the advantages of using fusion proteins.
We have not mentioned the use of siRNA as a therapeutic approach for ROR1-expressing malignancies.